# Factors Contributing to Non-Adherence to Treatment Among Adult Patients with Long-Term Haemodialysis: An Integrative Review

**DOI:** 10.3390/nursrep15090314

**Published:** 2025-08-26

**Authors:** Khin Chan Myae Win, Huaqiong Zhou, Vicki Patton, Mary Steen, Phillip Della

**Affiliations:** 1Division of Paediatric Intensive Care Unit, KK Women’s and Children’s Hospital, 100 Bukit Timah Road, Singapore 229899, Singapore; khin.chan.m.w@kkh.com.sg; 2Curtin School of Nursing, Curtin University, Kent Street, Bentley, Perth, WA 6845, Australia; h.zhou@curtin.edu.au (H.Z.); vicki.patton@curtin.edu.au (V.P.); mary.steen@curtin.edu.au (M.S.)

**Keywords:** dietary allowance, fluid allowance, haemodialysis, integrative review, non-adherence, routine haemodialysis

## Abstract

**Background:** Adherence of renal patients to a prescribed therapeutic regimen is crucial for the success of haemodialysis and the decreased mortality rates of patients; however, 60% are non-adherent to dialysis, fluid, and dietary allowances. To identify promising interventions aimed at improving treatment adherence, this review aimed to collate research evidence on the prevalence of non-adherence to treatment (fluid, diet, and routine haemodialysis) and to synthesise the factors contributing to non-adherence in long-term haemodialysis patients. **Methods:** An integrative review was conducted using Whittemore and Knafl’s five-stage framework (2005). ProQuest, CINAHL, PubMed, and Web of Science were searched, using the keywords ‘haemodialysis’ and ‘non-adherence’. The review included peer-reviewed quantitative studies published in English from 1 August 2018 to 30 June 2025, focusing on adults over 18 undergoing haemodialysis. The Joanna Briggs Institute (JBI) Critical Appraisal Checklist was used to assess the quality of the studies. **Results:** Twenty-nine studies were included, identifying factors across four treatment groups: non-adherence to fluid allowance, dietary allowance, haemodialysis session, and fluid/diet/haemodialysis. These factors were grouped into three themes: social demographics, clinical factors, self-management, and perceptions. Commonly cited factors included age, gender, educational status, health literacy, and perception. **Conclusions:** This review highlights the complex factors influencing treatment non-adherence, which may vary based on the variables and measurement tools used in each study. Low-level health literacy is the most frequently cited modifiable factor. Therefore, prioritising effective patient education that enhances knowledge and understanding of the importance of adhering to treatment is key to improving compliance in long-term haemodialysis patients.

## 1. Introduction

Chronic kidney disease (CKD) is the sixth fastest growing cause of death worldwide, accounting for approximately 2.4 million fatalities annually [1]. Driven by rising rates of obesity, diabetes, hypertension and an ageing population, CKD has become a global health burden with a significant economic burden [2]. Singapore ranks sixth in the world in terms of renal failure prevalence, with over 300,000 patients diagnosed with CKD and many more who remain undiagnosed [3].

CKD is classified into five stages based on the progression of the disease, measured by blood test results and symptoms. Stage 1 is defined as mild kidney damage with a normal estimated glomerular filtration rate (eGFR) of 90 or greater, and stage 5 is characterised as end-stage renal disease (ESRD) with an eGFR less than 15 [4]. The ESRD requires life-long renal replacement therapies such as haemodialysis, peritoneal dialysis, or kidney transplantation [5]. Haemodialysis filters the blood and removes waste, excess electrolytes, and fluids from the body by using dialysis machines [3]. Therefore, people with CKD are frequently required to strictly monitor and reduce their dietary fluid and mineral intake, including sodium, potassium, calcium, and phosphate [6]. Failure to adhere to these dietary restrictions is common among patients. Globally, among haemodialysis patients, the prevalence of non-adherence to dietary and fluid intake allowances is about 60% [7].

The World Health Organisation defines adherence to long-term therapy as “the extent to which a person’s behaviour—taking medication, following a diet, and/or executing lifestyle changes, corresponds with agreed recommendations from a health care provider” [8]. In the case of CKD, the patient’s commitment to the treatment plan is critical in predicting successful treatment and minimising complications such as deterioration, an increased admission rate, and inappropriate response to dialysis treatment [9]. Non-adherence to treatment, such as dietary and fluid allowances and routine dialysis, is a commonly recognised problem among patients undergoing long-term haemodialysis [9,10]. Non-adherence to treatment accelerates the progression of kidney disease into kidney failure [10], leading to cardiovascular mortality and even premature death [2]. Non-adherence to the regimen is, therefore, the leading cause of treatment failure and poor health outcomes in patients with ESRD [11].

Although there is no cure for renal failure, it is still possible to live a long and happy life with the correct care and therapy [3]. Adherence to the treatment regimen is crucial for achieving effective treatment and slowing the progression of the illness, thereby reducing complications and improving quality of life [12]. Therefore, it is important to identify reasons for non-adherence to treatment [13].

A considerable volume of studies has examined the non-adherence of dialysis patients [14,15,16,17,18]. However, the definition of “non-adherence” is broad and complex, making comparisons between outcomes difficult. Previous studies have reported diet and fluid non-adherence, while others have reported non-adherence to routine dialysis, medication, or lifestyle. Moreover, results were inconsistent among the studies. Therefore, it is necessary to systematically review the contributing elements of non-adherence to identify potential beneficial interventions to improve patients’ adherence to treatment for CKD [19].

This integrative review aimed to collate research evidence on the prevalence of non-adherence to treatment (fluid, diet, and routine haemodialysis) and to synthesise factors contributing to non-adherence to treatment among patients undergoing long-term haemodialysis.

### Research Question

The research question of this review was “What are the factors related to non-adherence to treatment (diet, fluid, and haemodialysis) among the adult patients with long-term haemodialysis?”

## 2. Materials and Methods

### 2.1. Design

An integrative review was conducted, guided by Whittemore and Knafl’s five-stage framework: problem identification, the literature search, data evaluation, data analysis, and presentation [20]. This review followed the Preferred Reporting Items for Systematic Reviews and Meta-Analyses (PRISMA) guidelines for the identification, screening, eligibility, and inclusion stages (Figure 1) [21].

### 2.2. Search Strategy

A comprehensive and systematic literature search was conducted on 1st August 2024 by searching four electronic databases: ProQuest, CINAHL Ultimate (EBSCO), PubMed, and Web of Science. The search was updated on the 28th and 29th of July 2025. Searches were conducted using advanced search functions in each database, applying the same or similar truncation methods after reviewing each database’s search guide. Key search terms were (Dialys* OR Hemodialysis OR Haemodialysis) and (nonadheren* OR non-adheren* OR “non adherent” OR “non adherence” OR non-complian* OR noncomplian* OR “non compliant” OR “non compliance”). The search terms were connected through the Boolean operators “AND” and “OR.” The preliminary search was conducted using specific limiters tailored to each database: in CINAHL, filters included publication within the last seven years (1 August 2018 to 30 June 2025), English language, research articles, peer-reviewed journals, and journal articles; in ProQuest, the search was limited to the last seven years (1 August 2018 to 30 June 2025), peer-reviewed, English language, scholarly journals, and articles; in PubMed, the search was restricted to free full-text articles published in the last seven years (1 August 2018 to 30 June 2025); and in Web of Science, filters included the English language, article type, and publication within the last seven years (2018–2025). The detailed search strategy and screening process were presented in the PRISMA flow diagram (Figure 1).

### 2.3. Inclusion and Exclusion Criteria

Articles eligible for inclusion are as follows: (1) primary quantitative studies and quantitative data extracted from mixed-method studies; (2) published in English; (3) included only adult patients (>18 years with long-term routine haemodialysis; (4) peer-reviewed journal articles from 1 August 2018 to 30 June 2025; (5) those focused on non-adherence to routine haemodialysis, fluid allowance, and dietary allowance, and (6) free full-text availability. Excluded studies were those that focused on qualitative research, those focusing on interventions related to treatment adherence, medication non-adherence, lifestyle, or other treatment types, such as peritoneal dialysis, short-term haemodialysis, or continuous renal replacement therapy. Reasons for exclusion were that previous studies have already evaluated medication non-adherence in dialysis patients, and non-adherence related to lifestyle and follow-up appointments are less concerning factors as compared to fluid, diet, and haemodialysis non-adherence. The optimal outcome in data analysis is promoted when an integrative review is conducted, incorporating similar study designs [20]. Therefore, only quantitative data, including data extracted from mixed-method studies, were included in this review to ensure the quality and consistency of quantitative findings.

### 2.4. Data Evaluation

Due to the nature of the question in this integrative review, the majority of risk factor-related studies are cross-sectional studies. The significant quantitative results from selected studies pertaining to the project objectives were included in this integrative review. The critical appraisal tools from the Joanna Briggs Institute Critical Appraisal Checklist (JBI) for analytical cross-sectional studies, case–control studies, and cohort studies were used to evaluate the studies (Table 1, Table 2 and Table 3) [22]. Following a thorough review of each study, its validity, potential bias, relevance, and reliability were evaluated using structured checklists specific to cross-sectional, case–control, and cohort study designs. This evaluation was carried out by two authors (KCMW and HZ).

### 2.5. Quality Appraisal

The risk of bias was generally low when the included studies were appraised using the specific question sets of the JBI checklist (Table 1, Table 2 and Table 3). Most studies clearly described their inclusion and exclusion criteria and provided adequate information about the study subjects and settings. Most studies used standardised measurements for the factors, and methods of statistical analysis were appropriate for each study. Nevertheless, some studies did not provide a sufficient discussion of tool validity and reliability nor details about the data collectors. There were a few studies that did not provide details about the study setting, the study timeframe, or the exclusion criteria.

### 2.6. Measurements and Examined Variables/Confounding Factors

There are two types of measurements used to identify factors associated with non-adherence in patients with CKD. The first method is patients’ self-reporting, and the second is based on biochemical parameters and intra-dialytic weight gain (IDWG) from medical records. However, the cut-off values for biochemical parameters and IDWG varied among the included studies. Therefore, it is important to note that differences exist in the indicators and standards used to assess treatment adherence across studies. Several previous studies have utilised serum potassium, albumin, and haemoglobin as indicators of dietary adherence, while others have employed only serum potassium and phosphate levels to assess dietary adherence [13,40].

### 2.7. Data Analysis

Thematic analysis was used to code the non-adherence factors by using the mind-mapping exercise by Whittemore and Knafl [20]. Subsequently, codes were organised under common themes, namely social-demographic factors, clinical factors, and factors related to patients’ self-management and perception of treatment adherence (Figure 2).

Included studies examined a variety of variables and confounding factors across the studies. Variables to evaluate different components of treatment (fluid, diet, or haemodialysis) varied across the studies. In the context of this review, “treatment” refers to diet, fluid, and haemodialysis; however, not all included studies examined all three components. Some studies reported adherence to only one component. Some assessed overall treatment adherence rather than reporting individual results separately for each component, even though the tools used measured adherence to diet, fluid, and haemodialysis. Therefore, the results are grouped based on the examined variables from the studies, namely:(1)Factors associated with non-adherence to fluid allowance, Table A1 (Appendix A);(2)Factors associated with non-adherence to dietary allowance, Table A2 (Appendix A);(3)Factors associated with non-adherence to haemodialysis, Table A3 (Appendix A);(4)Factors associated with non-adherence to fluid/diet allowances and haemodialysis, Table A4 (Appendix A).

## 3. Results

### 3.1. Search Outcome

The preliminary electronic database search yielded 3270 records after applying limiters to each database (Figure 1). After removing 312 duplicates using EndNote online, 2958 records remained. The titles and abstracts of 2958 records were screened for their relevance to the proposed objectives, and 2877 records were removed due to irrelevance. The full text of the remaining 81 potential articles was assessed, and 44 records were excluded because they did not address the proposed objectives or were unavailable in full text. The remaining 37 articles were evaluated against the predefined inclusion and exclusion criteria. Eight articles were further excluded because they were not primary studies; they included patients under the age of 18, as well as studies that focused on tool evaluation and the evaluation of educational interventions rather than treatment non-adherence/adherence. Finally, 29 articles were included in this integrative review.

### 3.2. Characteristics of Included Studies

Characteristics of the 29 included studies are presented in Table 4. These studies were conducted in different countries: Africa (two studies), Australia, Brazil, China, Egypt, Greece, Indonesia (two studies), Iran, Iraq, Israel, Japan, Korea, Malaysia, Pakistan, Saudi Arabia (three studies), Singapore, Slovakia (two studies), Turkey (three studies), the United States of America (three studies), Vietnam, and Yemen. One study was conducted in both Japan and the United States of America [42]. There were twenty-five cross-sectional studies, two mixed-method studies, one case–control study, and one cohort study. Data was collected via surveys, structured interviews, the extraction of laboratory results, and medical records. The sample size ranged from 41 [43] to 972 patients [16]. Most of the studies reported the odds ratio (OR) with 95% confidence intervals (CI).

### 3.3. Prevalence of Non-Adherence

Twenty included studies examined the prevalence of non-adherence to at least one component of treatment (fluid, diet, and haemodialysis) [17,24,25,26,29,30,31,32,33,34,35,37,38,39,40,41,42,43,44,45]. In terms of fluid allowance, non-adherence ranged from 38.4% to 62% across ten studies [24,26,30,33,34,38,39,40,41,44]. The prevalence of non-adherence to dietary restrictions was reported by eight studies, ranging from 39.1% to 63.7% [24,26,33,37,38,39,40,41]. The prevalence of non-adherence to haemodialysis ranged from 10% to 44.04% across eleven studies [24,25,26,29,32,33,35,40,42,43,45]. Three studies reported the overall non-adherence level, with prevalence ranging from 22.2% to 51.49% [17,24,26]. It is worth noting that only one study examined the prevalence of non-adherence to haemodialysis before and during the COVID-19 pandemic [18]. The study demonstrated that during the COVID-19 pandemic, significantly more patients were non-adherent to haemodialysis sessions compared to before the pandemic (19.5% vs. 11.7%; *p* < 0.01) [18].

### 3.4. Significant Factors Related to Non-Adherence to Fluid Allowance

Eight studies identified factors related to non-adherence to fluid allowance [27,30,34,38,40,41,44,46]. One study reported an inconclusive finding [14], and another study reported only the prevalence of fluid non-adherence [39]. The identified factors are categorised into three sub-themes such as social-demographic factors [gender, health literacy, educational status, age, and marital status], clinical factors (amount of urine output, depression), and factors related to the patient’s self-management and perception of adherence (self-efficacy, self-reported fluid non-adherence, reported difficulty with following fluid allowance, patient’s disease perception, attitude, and non-adherence behaviour towards fluid control) (Figure 2).

#### 3.4.1. Social-Demographic Factors

Health literacy was cited in two studies [38,41]. Skoumalova et al. [41] assessed patients’ health literacy by using a multi-dimensional tool consisting of nine health literacy questions. Patients with poor health literacy were less likely to engage actively with healthcare professionals, were more likely to be overhydrated (OR = 0.78, 95% CI: 0.65–0.94), and were less likely to self-report their fluid non-adherence behaviour (OR = 1.31, 95% CI: 1.07–1.59) [41]. Similarly, Indino et al. [38] identified that an overall higher health literacy is associated with increased fluid adherence (OR = 4.92, 95% CI: 1.13–21.35, and *p* = 0.033).

Three studies reported that patients’ educational status was a significant predictor, but the results were inconsistent [30,34,40]. According to Ozen et al. [40], patients who had completed only high school had a greater non-adherence rate to fluid allowance, compared to those with a higher education (OR = 4.377, 95% CI 1.502–12.75, and *p* = 0.007); likewise, Zhang et al. [30] reported that people with higher academic levels (college and above) were associated with a higher likelihood of adhering to fluid control (*p* = <0.001). In contrast, Perdana and Yen [34] found that patients with a higher education had a higher IDWG, which was an indicator of fluid non-adherence (*p* = 0.03).

Regarding age, older individuals showed greater fluid adherence (adjusted odds ratio [AOR] = 1.08, 95% CI = 1.02–1.14) [44], and married, as well as divorced/widowed, patients managed their fluid intake better than single patients (*p* = 0.029, *p* = 0.031) [30].

Moreover, males were also found to have a higher weight gain (higher IDWG score) when compared to females (*p* = 0.03, *p* < 0.001) [30,34]. However, Çankaya and Vicdan [27] found that male patients and patients who limited their fluid intake to ≤2000 cc between dialysis sessions had reported higher fluid adherence scores.

#### 3.4.2. Clinical Factors

Patients with a reduced daily urine output (<100 mL per day) were significantly associated with non-adherence to fluid control compared with adherent patients (*p* = 0.04) [34]. Similarly, Zhang et al. [30] reported that patients with a urine output of 100–400 mL or more than 400 mL in 24 h had a lower IDWG than those with a urine output of less than 100 mL in 24 h (*p* < 0.001). In addition, people with depression were found to be less adherent to their fluid restrictions (AOR = 0.82, 95% CI = 0.67–0.99) [44].

#### 3.4.3. Self-Management and Perception Factors

Perdana and Yen [34] assessed patients’ self-efficacy using the Indonesian Fluid Intake Appraisal Inventory (I-FIAI), with higher scores indicating greater self-efficacy [34]. The IFIAI instrument includes physiological, affective, social, and environmental components [34]. Patients with higher total self-efficacy scores had a lower IDWG (*p* = 0.05). However, non-adherent patients scored higher on self-efficacy related to physiological aspects, such as sensations of thirst and salty taste [34]. This means that non-adherent patients are reported to be more confident in their ability to avoid drinking due to thirst and dry mouth (*p* = 0.02) [34].

In another study, patients who reported difficulties adhering to the recommended fluid allowance were more likely to have a higher mean IDWG than those who did not report any difficulties (OR = 1.62, 95% CI: 1.08–2.43, and *p* = 0.02) [46]. In terms of perception and attitude, patients with greater symptom recognition had a more negative view of their illness (*p* < 0.001), while those with poor disease awareness had a negative attitude toward fluid management (*p* = 0.001) [30]. One study reported high scores in behaviour and attitude on the Fluid Control in Haemodialysis Patients Scale (FCHPS), which indicated that patients who adhere to fluid control actively follow fluid restrictions (*p* = 0.003) and have a positive mindset toward managing their fluid intake (*p* = 0.000) [27].

### 3.5. Significant Factors Related to Non-Adherence to Dietary Allowance

Nine out of twelve studies examined the factors associated with dietary allowance [14,27,36,37,38,39,40,41,46]. Two of the studies were inconclusive [14,46]. The significant factors related to dietary non-adherence included social-demographic factors [gender, educational status, body mass index (BMI), and health literacy], clinical factors [depression, anxiety], and factors related to patients’ self-management and perception of non-adherence (perceived benefit and barriers, perceived self-efficacy, self-management skills, flexibility in diet, difficulty in following recommended diet, attitude, and non-adherence behaviour about fluid control) (Figure 2).

#### 3.5.1. Social-Demographic Factors

Health literacy is the most frequently cited factor for dietary non-adherence, and all four studies reported consistent findings [36,37,38,41]. Dietary non-adherence was associated with a low nutrition literacy (r = 0.325, *p* < 0.001) and limited dietary knowledge (r = 0.361, *p* < 0.001) [36]. Skoumalova et al. [41] reported that lower scores of health literacy were indicators of dietary non-adherence (OR = 0.7 to 0.77); specifically, Skoumalova et al. [37] reported that patients with a moderate health literacy were more likely to be non-adherent to diet than patients with a higher health literacy (OR = 2.19; 95% CI: 1.21–3.99). Similarly, Indino et al. [38] also reported that patients with higher health literacy scores (in overall score and communicative health literacy score) adhered to the dietary allowance better than those with a lower health literacy (OR = 3.66, 95% CI 1.08–12.43, and *p* = 0.038).

Compared to people with higher educational backgrounds, high school graduates had a higher rate of non-adherence to dietary allowance (OR = 4.377, 95% CI: 1.502–12.75, and *p* = 0.007) [40], and non-adherence was found to be more common in patients with a higher BMI (OR = 1.06, 95% CI 1.00–1.13, and *p* = 0.030) [39]. In terms of gender, female patients demonstrated significantly higher mean dietary adherence scores compared to male patients (*p* < 0.001) [36]. This finding contrasts with the results of Çankaya and Vicdan [27], who reported that male patients exhibited a greater adherence to dietary recommendations than female patients.

#### 3.5.2. Clinical Factors

Moreover, patients with moderate/severe symptoms of anxiety (OR = 1.81; 95% CI: 1.22–2.69, and *p* < 0.01) and moderate/severe symptoms of depression (OR = 1.94; 95% CI: 1.26–2.98, and *p* < 0.01) were also more likely to be non-adherent to their diet control [37].

#### 3.5.3. Self-Management and Perception Factors

Patients who perceived fewer barriers (r = −0.369, *p* < 0.001) and higher benefits to follow diet recommendations (r = 0.246, *p* < 0.001) demonstrated a higher adherence [36]. Those patients who adhered to the fluid allowance perceived the importance of limiting fluid intake and were also associated with a better dietary adherence (OR = 9.42, 95% CI 5.03–17.63, and *p* < 0.01 and OR 1.90, 95% CI 1.15–3.14, and *p* = 0.0121) [39]. In addition, a positive correlation between the diet subscale score of the ESRD-AQ and the attitude and behaviour scores of the FCHPS indicates that patients exhibiting more favourable attitudes and behaviours toward fluid control are also more likely to demonstrate a higher adherence to dietary recommendations (*p* = 0.000) [27].

Moreover, Opiyo et al. [39] reported that flexibility in the diet options (aOR = 2.65, 95% CI 1.11–6.30, and *p* = 0.028) promoted adherence, whereas reported difficulties in following diet recommendations (aOR = 0.24, 95% CI 0.13–0.46, and *p* < 001) and adherence to limiting fluid intake (aOR 9.74, 95% CI 4.90–19.38, and *p* < 0.001) were the inhibiting factors. Dietary adherence was also reported to be significantly linked to specific components of health beliefs, specifically the perceived self-efficacy skills (r = 0.550, *p* < 0.001) and self-management skills (r = 0.465, *p* < 0.001) [36].

### 3.6. Significant Factors Related to Non-Adherence to Routine Haemodialysis

Twelve studies identified factors associated with non-adherence to routine haemodialysis [14,16,27,29,31,32,35,40,42,43,45,46]. These significant factors included social-demographic factors (male, marital status, those with children, poor knowledge, age, religion, personal transportation, busy lifestyle, smokers, dialysis schedule, black patients, educational status, haemodialysis vintage, and patients on a transplant waiting list), clinical factors (having a central venous catheter, day of dialysis, experiencing difficulties during procedure, last advice given by healthcare professionals, and frequency of advice given), and factors related to patients’ self-management and perception of treatment adherence included the perceived importance of haemodialysis, patients’ general procrastination, nurse’s attitude, motivation, coping, and perceived family and health worker support (Figure 2).

#### 3.6.1. Social-Demographic Factors

Patients with no children (*p* = 0.005) [27], those who used their private transportation to the dialysis centre (OR = 1.82, 95% CI 1.05–3.15, and *p* < 0.05) [32], who identified as Christian (*p* = 0.003) [43], who were waiting for a kidney transplant (OR = 0.24; 95% CI 0.083–0.72, and *p* = 0.01) [42] and who were high school graduates (OR = 0.20, 95% CI 0.05–0.81, and *p* = 0.02) [42] were statistically significantly associated with adherence to haemodialysis. In contrast, non-adherence to haemodialysis was more common in black patients (OR = 3.98, 95% CI 1.42–11.22, and *p* < 0.001) [42], male patients (*p* = 0.05) [45], (OR = 2.074, 95% CI 1.213–3.546, and *p* = 0.008) [40], smokers (*p* = 0.003) [45], those patients whose dialyses were scheduled during night instead of daytime (*p* = 0.042) [45], and those who claimed to lead a busy lifestyle (OR = 0.59, 95% CI 0.35–0.98, and *p* < 0.05) [32].

Two studies reported that married patients demonstrated a higher adherence to routine haemodialysis (*p* < 0.05) (OR = 2.11, 95% CI 1.23–3.60, *p* < 0.01) [27,32]. In terms of age, Dantas et al. [35] reported a significant negative correlation between age and IDWG (r = −0.41; *p* < 0.001) and a reduction in sessions (r = −0.31; *p* = 0.005); however, the specific age range was not clearly specified in the study. Mukakarangwa et al. [43] reported that age (range, 41–50 years) was statistically significantly associated with adherence to haemodialysis (95% CI, 26.76–29.17; *p* = 0.038). Therefore, age appears to influence haemodialysis adherence and related outcomes, though differences in age reporting limited the direct comparison.

Regarding haemodialysis vintage, Le et al. [16] reported that patients with more than 5 years of haemodialysis were less likely to adhere to haemodialysis (*p* < 0.001). In contrast, Ozen et al. [40] reported that patients with longer experience were more adherent (OR = 0.992, 95% CI 0.986–0.998, and *p* = 0.005), though the specific duration was not stated. Additionally, two studies reported that greater knowledge positively influenced adherence to the haemodialysis programme (*p* = 0.005) (t = 2.234, *p* = 0.028) [27,29].

#### 3.6.2. Clinical Factors

Patients were non-adherent to haemodialysis when they suffered from depression (t = −4.190, *p* = 0.001) [29] and experienced difficulties during the dialysis procedures, such as low blood pressure, muscle spasms, pain, and headache (95% CI 20.80–28.36, *p* = 0.004) [43]. Patients scheduled for dialysis on Sundays, Tuesdays, and Thursdays were more likely to attend than those scheduled on Saturdays, Mondays, and Wednesdays (OR = 1.65, 95% CI 1.03–2.64, and *p* < 0.05) [32]. Furthermore, patients with central venous catheters were more than twice as likely to be non-adherent (OR = 2.591, 95% CI 1.171–5.733, and *p* = 0.019) [40]. It was also found that how long patients had the kidney disease and whether they had other chronic illnesses affected how well they adapted to haemodialysis and participated in haemodialysis (*p* = 0.005), but it does not explicitly say whether having a chronic disease leads to a better or worse adherence [27].

In terms of patient education, the greater the frequency of education given by healthcare professionals, the better the rate of adherence to dialysis (OR = 0.45, 95% CI 0.21–0.97, and *p* < 0.05) [32], (*p* < 0.01) [43]. Patients who received education one month before participating in the study were less likely to be adherent than those who received the advice the same week (OR = 0.41, 95% CI 0.18–0.93, and *p* < 0.05) [32].

#### 3.6.3. Self-Management and Perception Factors

Patients’ perception of the importance of haemodialysis was significantly correlated with adherence to haemodialysis (OR = 2.62, 95% CI 1.03–6.67 and 95% CI 20.44–27.76, *p* = 0.020) [32,43]. It is also worth noting that those who self-reported difficulties coming to dialysis were found to have a higher absenteeism for dialysis sessions [46]. However, this was not statistically significant (OR = 1.41, 95% CI 0.96–2.07, and *p* = 0.09) [46]. A similar result was reported, showing that higher levels of motivation, coping ability, perceived family support, and perceived health worker support were all significantly associated with a better adherence to the haemodialysis programme (*p* = 0.001) [29].

Dialysis attendance was noted to have a significant inverse relationship with general procrastination (*p* = 0.01) [14]. Procrastination is defined as the failure in self-regulation that manifests as delayed choice and decision that persists over time, despite repeated opportunities to make the change [14]. Interestingly, patients’ adherence was negatively related to nurses’ attitudes toward participation; that is, nurses who strongly believed patients should be involved in decisions were more likely to accommodate the patient’s wish to shorten haemodialysis time (β = −0.07, *p* < 0.05) [31].

### 3.7. Significant Factors Related to Non-Adherence to Fluid/Diet Allowances and Haemodialysis

This theme encompasses the general findings of non-adherence from eleven studies [14,16,17,18,23,24,25,26,27,28,33]. One study reported an inconclusive finding [15]. The significant predictors related to non-adherence to treatment were social-demographic factors (gender, older age, education status, medication payment ability, employment status, social status, social support, rural residency, low health literacy, dietary knowledge, longer haemodialysis vintage, early onset of dialysis, and a three dialysis weekly schedule), clinical factors (hypertension, history of COVID-19, fear of COVID-19, having suspected COVID-19 symptoms, depressive disorder, psychiatric treatment history, and low SMME score), and self-management and perception factors (perception score and understanding score, low perceived social support, frequent complaints, poor self-care behaviour, low harm avoidance, high reward dependency, and irregular or less frequent self-care behaviour education) (Figure 2).

#### 3.7.1. Social-Demographic Factors

Five studies reported the same finding that older people were correlated with adherence to treatment as compared to younger people [14,16,17,25,26]. Le et al. [16] defined older age as between 60 and 85 years, Raashid et al. [17] defined older age as people with a mean age of 51.05 ± 13.80 years, Erkan et al. [25] defined it as 54.5 ± 14.8 years, and Alatawi et al. [26] defined it as 60 years or older, respectively. Bazrafshan et al. [14] did not clearly specify the age range. In addition, Chan et al. [28] reported an association between age and treatment non-adherence (diet/fluid); however, they did not specify whether this applied to older or younger patients, nor did they present the results clearly. One study claimed that patients who had started haemodialysis at a younger age are more likely to be non-adherent to haemodialysis (*p* = 0.038) [25]. Regarding gender, two studies reported contrasting results, with one finding higher adherence scores in females (*p* = 0.008, r = 0.327) [23] and the other in males (*p* < 0.05) [27].

In addition, unemployed patients (*p* = 0.050) [26], patients with urban residency (*p* = 0.03) [24], those with a “very or fairly easy” ability to pay for prescribed medications (B = 27.92, 95% CI 5.89–49.95, and *p* = 0.013) [16], and those with two dialysis weekly schedules (*p* < 0.001, r = −0.411) [23] are more likely to comply with treatment [16]. Moreover, two studies found that a higher social status was associated with better overall treatment adherence (B = 13.87, 95% CI: 0.25–27.49, and *p* = 0.046; *p* = 0.023) [16,33]. The patients with a higher education (F = 9.97, *p* < 0.001) [28,33] and those with a higher digital healthy diet literacy (B = 1.35, 95% CI 0.59–2.12, and *p* = 0.001) [16] had higher adherence scores for overall treatment. However, regarding treatment non-adherence, patients on longer haemodialysis vintage (>5 years) were less likely to adhere to overall treatment (B = −52.87, 95% CI, −70.46–−35.28, and *p* < 0.001) and fluid/diet recommendations (B = −18.70, 95% CI −30.53–−6.87, and *p* = 0.002) [16]. Two other studies demonstrated the same result that patients with a shorter haemodialysis duration had significantly higher adherence levels than those with longer durations (949.2 vs. 908.0, *p* = 0.02) (r = −0.386, *p* = 0.002) [24,25]. Surprisingly, in one study, people with poor adherence to fluid and dietary allowances had better haemodialysis dietary knowledge (B −4.92; 95% CI −7.51–−2.34, and *p* = 0.001) [16].

#### 3.7.2. Clinical Factors

Patients with renal disease related to hypertension showed poorer treatment adherence than those with other types of renal disease (*p* = 0.02) [14]. High systolic blood pressure (*p* = 0.047), the type of depressive disorder (current recurrent) (*p* = 0.049), having psychiatric treatment history (*p* = 0.031), and a lower SMME score (more severe cognitive impairment) (r = −0.295, *p* = 0.021) were the risk factors for non-compliance to treatment [25].

Among the included studies, two studies reported that the COVID-19 pandemic was a concerning factor for dialysis patients’ adherence [16,18]. Patients with a COVID-19 history were 5.36 times more non-adherent than those without a COVID-19 history (OR = 5.36, 95% CI 1.79–16.1, and *p* = 0.003) [18]. Non-adherent patients showed higher scores for fears of COVID-19 when compared to adherent patients, implying that non-adherent patients are more terrified of COVID-19, which led to non-adherence. (OR = 1.06, 95% CI 1.01–1.11, and *p* = 0.029) [18]. Le et al. [16] reported the same result: patients who were more afraid of COVID-19 were less likely to adhere to both overall treatment (B = −1.78; 95% CI, −3.33–−0.24, and *p* = 0.023) and fluid/dietary allowances (B = −1.70, 95% CI −2.71–−0.68, and *p* = 0.001). However, patients with suspected COVID-19 symptoms were observed to be more adherent to fluid and dietary allowances (B = 27.13, 95% CI 10.78–43.49, and *p* = 0.001) [16].

#### 3.7.3. Self-Management and Perception Factors

Non-adherent patients had a poorer understanding of the importance of treatment adherence (OR = 0.05, 95% CI 0.01–0.29, and *p* = 0.001) [18]. Patients who presented with increased self-care behaviour (r = 0.62, *p* < 0.001) and those who attended at least two self-care behaviour education sessions per year (F = 4.22, *p* = 0.020) showed higher treatment adherence [33]. On the other hand, patients with more frequent subjective cognitive complaints exhibited poorer treatment adherence [28].

Additionally, the non-adherent group was reported to have lower perception scores (OR = 0.76, 95% CI 0.62–0.92, and *p* = 0.006), indicating poorer knowledge and attitudes about the importance of treatment adherence [18]. Similarly, patients with positive perceptions of overall treatment scores (*p* ≤ 0.001) [24], fluid restriction scores (*p* ≤ 0.001) [24], diet scores (*p* ≤ 0.001) [24], and social support (*p* = 0.019) [26] showed a significantly higher adherence to overall treatment. Low harm avoidance (*p* = 0.012) and high reward dependency (*p* = 0.014) were also the prominent temperament features of the non-compliant patients [25].

## 4. Discussion

This integrative review sought to synthesise the factors related to non-adherence to treatment, the prevalence of treatment non-adherence, and formulate recommendations on how to increase the adherence rate among patients undergoing long-term haemodialysis. This integrative review discusses, based on the commonly cited factors across the four treatment groups, different tools and indicators used to measure non-adherence and recommendations for future clinical practice.

A large variety of factors identified from 29 studies were grouped under “social-demographic factors,” “clinical factors,” and “self-management and perception factors”. Different risk factors and inconsistent findings were identified due to the heterogeneity of the studies, including the examined variables and the different tools used to measure adherence.

### 4.1. Most Frequently Cited Factors Across Four Groups

After analysing the extracted risk factors from the included studies, the most frequently cited factors across the four non-adherence themes and three influencing-factor themes are gender, age, educational status, poor health literacy, and perception of the importance of treatment adherence.

Age was a most frequently cited risk factor in this review, reported in nine studies; however, the age ranges examined and the definitions of “older” varied across studies, leading to inconsistent findings [14,16,17,25,26,28,35,43,44]. Generally, older age groups have been reported to be more likely to adhere to treatment [14,16,17,25,26,35,44]. This may be because as elderly individuals age, their oral intake decreases, leading to a better adherence to dietary and fluid allowances [17]. Similarly, a previous integrative review reported that younger patients were less adherent to dietary allowances [47]. Experiencing chronic disease at a young age can reduce self-control and motivation, with non-adherence sometimes used as a way to regain control [25]. It was supported by another systematic review that younger individuals often have more social relationships outside the home, increasing exposure to situations that challenge their dietary and fluid control [47]. Additionally, Chironda and Bhengu [19] also reported that middle-aged and younger patients often struggle with adherence due to family commitments. Additionally, younger individuals might perceive fewer immediate health consequences from non-adherence compared to older patients, which can reduce their motivation to comply [19].

Multiple studies reported conflicting findings regarding gender, with some studies showing higher adherence scores in females [23,30,34,36], while other studies found a greater adherence in males [27,45]. The inconsistent results may be partly attributed to the differing proportions of male and female participants in the studies, which could have affected the observed adherence patterns. Variations in gender distribution might have influenced the outcomes, leading to contrasting findings across the studies.

Six studies reported patients’ educational status as a significant predictor in this review [28,30,33,34,40,42]. Five studies identified that patients with a higher education were more likely to be compliant with treatment than those with a lower academic background [28,30,33,40,42]. The previous reviews have supported this finding that patients with low educational backgrounds are less likely to comprehend the information provided and are less likely to adhere to treatment management [9,19]. One of the included studies reported that patients with a lower academic status are more compliant than more educated patients, causing the review’s result to be inconsistent [34]. However, one factor contributing to inconsistent results in this review could be the measurement tool; that is, the authors only used the objective measurement, IDWG, to indicate that the patient is not adherent to treatment [34]. Using IDWG alone to determine non-adherence may not be adequate to conclude that educated patients are less adherent.

This review reported consistent findings that patients with a lower health literacy are more likely to be non-adherent to their treatment [16,36,37,38,41]. One study also found that patients with more extended dialysis had a higher dietary literacy, as their health literacy improved gradually over the years [36]. Additionally, patients with poor health literacy are vulnerable to receiving incorrect nutrition information because they lack the critical literacy skills to assess the relevance and accuracy of the content [47]. Generally, health literacy is interrelated with patients’ perceptions. A significant number of studies have reported the same finding that patients who perceive the importance of adherence are more likely to adhere to treatment plans [18,24,26,29,30,32,36,39,43]. This finding is consistent with a systematic review that reported patients’ beliefs and perceptions influence their ability to adjust to the demands of treatment [19]. A study conducted in Egypt similarly found that perception scores were a key predictor of non-adherence [18]. Improving patients’ perception of the importance of adhering to various treatment domains is therefore recommended to enhance overall treatment adherence.

### 4.2. Inconsistent Tools and Non-Adherence Indicators Across the Studies

One probable reason for the inconsistent findings of contributing factors is the measurement tools used in the included studies. Adherence was primarily measured using subjective self-reported adherence questionnaires [14,15,16,17,18,23,24,25,26,27,28,29,30,31,32,33,34,35,36,37,38,39,40,41,42,43,44,45,46]. Among the self-reported adherence questionnaires, the End-Stage Renal Disease Adherence Questionnaire (ESRD-AQ) is the most commonly used self-reported adherence tool, applied in eleven of the 29 studies [14,16,17,18,23,24,26,27,36,39,43]. Only a few studies examined both subjective adherence questions and objective laboratory indicators and medical records to determine non-adherence [25,28,30,34,35,45].

Among the self-reported questionnaires, the ESRD-AQ is the most common tool to measure patients’ treatment adherence behaviours, knowledge, and perception in four dimensions: haemodialysis attendance, medication compliance, fluid, and diet recommendations. The content validity index for this instrument is 0.99, and the test–retest correlation value is 0.915 [48]. There are several ways to assess how well people with *ESRD* adhere to their treatment plans, and there is no single “gold standard” for measuring non-adherence [49]. Therefore, authors have been using different methods and questionnaires to measure adherence over the past years. There are biological measures, such as IDWG, and biochemical indicators, like pre-haemodialysis serum potassium or phosphorus levels, to evaluate treatment non-adherence in patients receiving maintenance haemodialysis [13]. These indicators are used to assess not only treatment adherence but also clinical outcomes in patients undergoing haemodialysis [24]. There is no universally standardised cut-off value for each marker, and it is unclear if these measurements are a reliable method for assessing non-adherence rates in the haemodialysis population [49]. Fernandes and D’Souza [13] argued that these objective measurement methods may be more effective or reliable in measuring clinical outcomes but may not be adequate for measuring non-adherence. Lambert et al. [50] reported that a combination of both subjective information and laboratory measures is recommended in assessing patients’ adherence levels. Therefore, future studies should consider adopting both subjective and objective ways of collecting information to assess treatment adherence.

### 4.3. Implications and Recommendations

Healthcare professionals play a critical role in providing accurate information on management for dialysis patients; however, patient education to improve health literacy is often not their top concern due to time constraints [51]. Under half of the participants reported that they rarely received counselling/education from healthcare professionals [36]. Barriers to patient education may include a lack of motivation and inadequate support for decision-making [9]. An integrative review by Oquendo et al. [47] found that diet adherence is influenced by the relationship between patients and their healthcare professionals. However, when there are problems with communication or when insufficient time, information, or empathy is received from healthcare professionals, it can also inhibit treatment adherence [47].

A low level of health literacy can contribute to poor comprehension, leading to non-adherence and difficulties in following instructions [19]. For example, patients who perceive the benefits of diet modification are reported to be more likely to adhere to it compared to those who perceive diet restriction as a burden, according to one integrative review [50]. Studies have shown that there is still a lack of patient awareness regarding the prevention of CKD, the importance of early detection, and the significance of treatment adherence [52]. Although studies report poor health literacy and its relationship to treatment non-adherence, there are limited studies examining the patient’s perception and non-adherence. Therefore, more research is required.

Based on the findings from this review, healthcare professionals should pay closer attention and provide additional support when dealing with dialysis patients who are younger, less educated, and have a low health literacy. Considering all the factors discussed, healthcare professionals need to tailor any interventions based on patients’ particular risk factors, aiming to improve their perception of the importance of treatment adherence [31]. Healthcare professionals need to take an active role in screening patients for factors that could impact their adherence.

Effective teaching strategies should be formulated to enhance patients’ knowledge and health literacy about disease management. Among the various methods of health education, the teach-back method is particularly beneficial in educating individuals with diverse levels of health literacy [53,54]. This is because this method requires patients to repeat back the instructions given to ensure they fully understand them [36]. The teach-back method improves not only patients’ knowledge but also their self-management and self-efficacy in managing chronic diseases [55]. Recently, one randomised control trial reported that peer education can also significantly enhance adherence among haemodialysis patients for both attendance at sessions and fluid recommendations [56]. Further research should be conducted on effective health education and patients’ non-adherence to treatment among haemodialysis patients, as there is currently a paucity of evidence on this subject. Additionally, this review excluded qualitative studies, which may limit the exploration of patient perceptions, experiences, and narratives that could provide deeper contextual understanding. While the inclusion of quantitative studies allowed us to focus on measurable outcomes, qualitative research could offer valuable insights into the subjective dimensions of patient care. Future reviews incorporating qualitative syntheses are therefore recommended as a complementary approach to enrich and extend the findings of this review.

## 5. Limitations

The generalisability of findings from this integrative review is limited by several factors, including differences between developed and developing countries, various cultural contexts, available facilities, and minority groups, which may not reflect the broader dialysis population. This review’s focus on quantitative studies and quantitative data extracted from mixed-methods research may have limited the inclusion of relevant qualitative evidence, thereby overlooking important social, cultural, and experiential dimensions that qualitative data can provide. Additionally, although treatment adherence in this review is defined as adherence to diet, fluid, and haemodialysis, many studies reported adherence to only one of these components or provided an overall adherence measure rather than reporting each aspect separately. This inconsistency complicates efforts to generalise findings, interpret results comprehensively, and synthesise the data effectively.

The review included only published studies in English, potentially excluding relevant research published in languages other than English. Restricting the review to studies with free full-text availability may also have resulted in the exclusion of potentially valuable, inaccessible articles. Studies that did not explicitly focus on this review’s defined concept of treatment adherence but mentioned it briefly while addressing other topics (e.g., medication adherence or intervention for adherence) were excluded. As a result, potentially relevant data related to treatment adherence may have been missed. Lastly, this review presents COVID-19-related findings; however, these should be clearly framed as observations specific to the pandemic period, reflecting conditions and healthcare practices that may not persist beyond that time. Therefore, these findings are tied to the unique circumstances of the pandemic and may not be applicable outside the context of the pandemic.

## 6. Conclusions

To address the non-adherence issues (fluid, diet, and haemodialysis) among patients receiving haemodialysis, understanding the predisposing factors is imperative. This integrative review identified three main types of contributing factors, including “social-demographic factors”, “clinical factors” and “self-management and perception factors”. The most frequently cited factors are gender, older age, low educational background, poor health literacy, and perception of importance of treatment adherence. Healthcare professionals play a crucial role in identifying patients’ contributing factors and providing accurate and consistent information on disease management to enhance patients’ health literacy. Therefore, it is essential to tailor teaching strategies to each patient’s specific factors and deliver effective education on treatment adherence.

## Figures and Tables

**Figure 1 nursrep-15-00314-f001:**
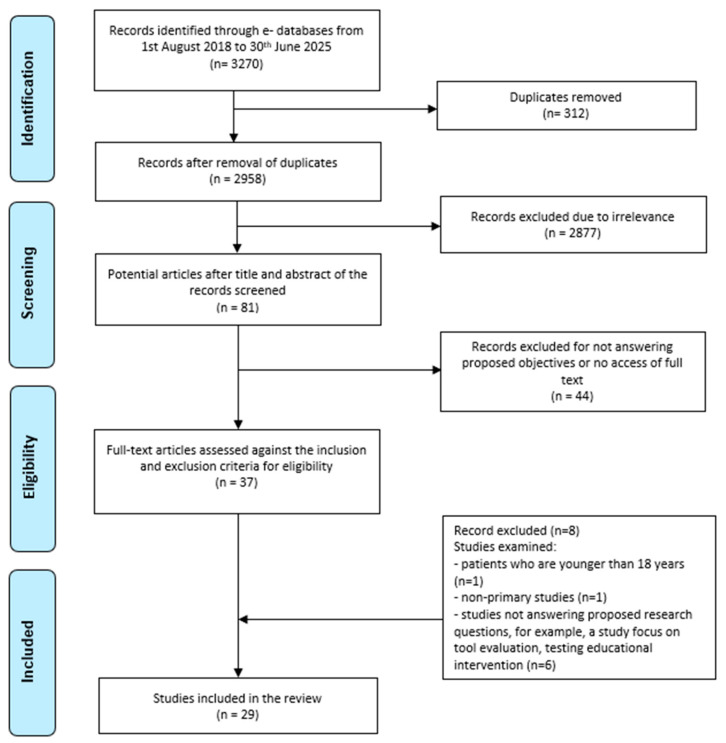
PRISMA flow diagram of search strategy results [21].

**Figure 2 nursrep-15-00314-f002:**
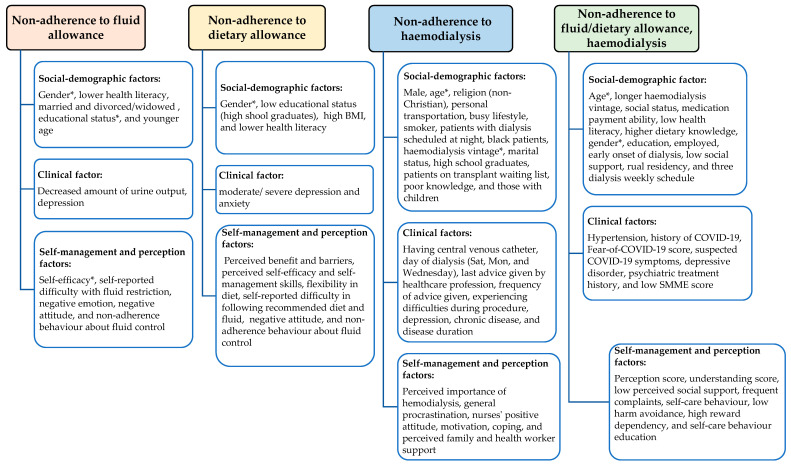
Conceptual map of factors under four variable themes. (*)—inconsistent findings from different studies.

**Table 1 nursrep-15-00314-t001:** Methodological critical appraisal using the JBI cross-sectional checklist.

References	Clearly Defined Inclusion Criteria in the Sample	Study Subjects and Settings Described in Detail	Exposure Measured in Valid and Reliable Way?	Objective, Standard Criteria Used for Measurement of Condition	Confounding Factors Identified	Strategies to Deal with Confounding Factors Stated?	Outcomes Measured in a Valid and Reliable Way	Appropriate Statistical Analysis Used
Ansaf & Al-Hamadani (2025) [23]	✓	✓	✓	✓	✓	X	✓	✓
Belhmer et al. (2025) [24]	✓	✓	✓	✓	X	X	✓	X
Erkan et al. (2025) [25]	✓	✓	✓	✓	✓	X	✓	✓
Alatawi et al. (2024) [26]	✓	✓	✓	✓	X	X	✓	✓
Çankaya & Vicdan (2024) [27]	✓	✓	✓	✓	X	X	✓	✓
Chan et al. (2024) [28]	✓	✓	✓	✓	✓	✓	✓	✓
Rondhianto et al. (2024) [29]	✓	✓	✓	✓	✓	✓	✓	✓
Bazrafshan et al. (2023) [14]	✓	✓	✓	✓	X	X	✓	✓
Le et al. (2023) [16]	✓	✓	X	✓	✓	✓	✓	✓
Zhang et al. (2023) [30]	✓	✓	✓	✓	✓	X	✓	✓
Fotaraki et al. (2022) [15]	✓	✓	✓	✓	✓	✓	✓	✓
Idilbi et al. (2022) [31]	✓	✓	✓	✓	X	X	✓	✓
Sultan et al. (2022) [18]	✓	✓	✓	✓	✓	✓	X	✓
Alzahrani et al. (2021) [32]	X	X	✓	✓	✓	✓	✓	✓
Kim & Cho. (2021) [33]	X	✓	X	X	✓	X	✓	✓
Perdana & Yen (2021) [34]	✓	X	✓	✓	✓	✓	✓	✓
Raashid et al. (2021) [17]	✓	✓	X	✓	✓	✓	✓	✓
Dantas et al. (2020) [35]	✓	✓	✓	✓	✓	X	✓	✓
Lim et al. (2020) [36]	✓	✓	✓	✓	✓	✓	✓	✓
Skoumalova et al. (2020) [37]	✓	✓	✓	✓	✓	✓	X	✓
Indino et al. (2019) [38]	✓	✓	✓	✓	✓	✓	X	✓
Opiyo et al. (2019) [39]	✓	✓	X	✓	✓	✓	✓	✓
Ozen et al. (2019) [40]	x	✓	✓	✓	✓	✓	✓	✓
Skoumalova et al. (2019) [41]	✓	✓	X	✓	✓	✓	✓	✓
Miyata et al. (2018) [42]	✓	✓	X	✓	✓	✓	X	✓
Mukakarangwa et al. (2018) [43]	✓	X	✓	✓	✓	✓	X	✓
Washington et al. (2018) [44]	✓	✓	X	X	✓	✓	✓	✓

**Table 2 nursrep-15-00314-t002:** Methodological critical appraisal using the JBI case–control checklist.

References	Were the Groups Comparable?	Were Cases and Controls Matched Appropriately?	Were Criteria for Identifying Cases/Controls Clearly Defined?	Was Exposure Measured in a Valid and Reliable Way?	Were Cases/ Controls Selected the Same Way?	Were Confounding Factors Identified?	Were Strategies to Deal with Confounding Factors Stated?	Was Outcome Assessed in a Valid and Reliable Way?	Was Exposure Period Long Enough?	Was Appropriate Statistical Analysis Used?
Alhawery et al. (2019) [45]	✓	✓	✓	✓	X	X	✓	✓	✓	✓

**Table 3 nursrep-15-00314-t003:** Methodological critical appraisal using the JBI cohort study checklist.

References	Groups Similar and From Same Population?	Exposure Measured Similarly?	Exposure Measured Validly and Reliably?	Confounders Identified?	Were Strategies to Deal with Confounding Factors Stated?	Participants Free of Outcome at Start?	Outcome Measured Validly and Reliably?	Follow-Up Time Sufficient?	Follow-Up Complete?	Strategies for Incomplete Follow-Up?	Appropriate Statistical Analysis?
Snyder et al. (2020) [46]	✓	✓	X	✓	✓	✓	✓	✓	X	X	✓

**Table 4 nursrep-15-00314-t004:** Characteristics of included studies.

References	Study Setting	Study Design/ Data Collection	Sample Size	Participants	Data Analysis
Ansaf & Al-Hamadani (2025) [23] Iraq	Dialysis Centre of Baghdad Medical City	Cross-sectional study Questionnaire	72	Mean age = 53.83 ± 13.52 Male:Female = 52.8%:47.2% Being married = 87.5% Illiterate = 12.5%, academic education = 30.6%, and primary and secondary school education = 56.9% Number of dialysis sessions per week: two sessions per week = 76.4% and three sessions per week = 23.6% Renal transplant history: no previous renal transplantation = 94.4% Primary cause of CKD: diabetic nephropathy = 37.5%, hypertension = 27.7%. Health literacy: average readability = 3.33 ± 1.583 ESRD-AQ average score = 782.29 ± 133.245	Multivariate linear regression analysis, the Kolmogorov–Smirnov test, and the Spearman correlation test
Belhmer et al. (2025) [24] Yemen	Urology and Nephrology Centre of a tertiary public hospital	Descriptive cross-sectional study Face-to-face interviews using a semi-structured questionnaire	393	Mean age = 45.0 ± 17.4, males = 62.9%, married = 81.9%, and lived in urban areas = 88.6% Unemployed or retired = 92.1%, illiterate = 43.0% Living with their families = 88.0% Undergoing haemodialysis for <5 years = 61.8% Hypertensives = 80%, both hypertension and diabetes mellitus = 16% Good perception toward the dialysis = 98.7% Lowest perception toward diet restriction = 85.8% Fair perception of importance of fluid restriction = 89.1% Received counselling by a healthcare provider = 99.0% Received counselling about the dialysis attendance = 17.6% Received counselling about compliance with medication = 34.6% Received counselling about diet recommendations = 30.8% Received counselling about fluid restriction = 31.0%	The Mann–Whitney U-test
Erkan et al. (2025) [25] Turkey	Karşıyaka Nephron Dialysis Centre and Ege Nephrology Dialysis Centre	Descriptive and cross-sectional study Diagnostic interview Laboratory measurements	61	Mean age = 51.5 ± 15 Men:Women = 62.3%:37.7% Married = 72.1%, single = 16.4%, divorced = 3.3%, and widowed = 8.2%	Shapiro–Wilk test, Mann–Whitney U test, cross-tabulations, Chi-square (χ2) analysis, and Spearman correlation analysis
Alatawi et al. (2024) [26] Saudi Arabia	Two haemodialysis units at two major hospitals	Quantitative cross-sectional correlational study Survey using questionnaire	121	18–40 years old = 31.4%, 41–50 years = 24%, and 51–60 years = 19.0% Male:Female: 56.2%:43.8% High school graduates = 38%, bachelor’s or higher degrees = 14% Married = 49.6%, single = 25.6%, and divorced = 24.8% Unemployed = 49.6%, employed = 21.5%, student = 6.6%, and retired = 22.3% Monthly income of <3000 SR = 55.4%, >3000 SR = 44.6%	Fisher’s exact test
Çankaya & Vicdan (2024) [27] Turkey	Haemodialysis unit of a public hospital	Cross-sectional study Questionnaire and face-to-face structure Interviews	71	Mean age = 46.94 ± 15.42 (18–75 years) Female:Male = 54.9%:45.1% Primary school or lower = 50.7% Employed = 52.1%, social security = 64.8% Married = 57.7%, with children = 57.7%, and 3 or more children = 70.7% Reported equal income to the expenses = 60.6% Living with their parents = 70.4% Duration of dialysis: 47.9% for 13 months to 5 years	Pearson’s correlation test Kolmogorov–Smirnov test Independent sample *t*-test One-way ANOVA The Bonferroni test The Tamhane’s T2 test The Levene test
Chan et al. (2024) [28] Singapore	Two haemodialysis units of two major hospitals	Cross-sectional study Survey	268	Mean age = 59.87 ± 11.72 (26–84 years) Female:Male = 42.5%:57.5% Mean years of education = 9.59 ± 3.56 years Mean months of duration of dialysis = 78.85 ± 62.8 months	Structural equation modelling
Rondhianto et al. (2024) [29] Indonesia	Haemodialysis unit of a community hospital	Cross-sectional study Survey	90	Mean age = 46.82 ± 12.98 (18–65 years) Female:Male = 63.33%:26.67% Elementary school = 33.33% Duration of dialysis = 6 months to 10 years Good knowledge = 93.33%, good coping = 91.11% Perceived family support and health worker support = 90% and 88.89% High motivation = 100%	Multiple linear regression test, Chi-square test
Bazrafshan et al. (2023) [14] Iran	Six dialysis centres	Descriptive correlational study Questionnaire	218 patients	Mean age = 54.11 ± 14.78 years Employed = 13.8%	Multivariate regression analysis) statistics
Le et al. (2023) [16] Vietnam	Eight hospitals	Cross-sectional study Survey	972 patients	Male = 53.46%, Female = 46.54%	T-test, one-way ANOVA test, and bivariate and multivariate linear regression models
Zhang et al. (2023) [30] China	Dialysis centres	Cross-sectional study Questionnaire Medical records	253 patients	Mean age = 48.83 (SD = 12.38 years) Male = 161 (63.64%) Female = 92 (36.36%)	Multivariate regression analysis, Pearson’s correlation, t-test, and one-way analysis of variance
Fotaraki et al. (2022) [15] Greece	Haemodialysis unit of bioclinic hospital	Cross-sessional study Questionnaires	100 patients	Males = 57 (57%) Females = 43 (43%) Median duration on dialysis = 4 years Married = 55% High school graduates = 54% >50 years old = 66%	Multivariate linear regression Spearman’s rho correlation coefficient
Idilbi et al. (2022) [31] Israel	Galilee Medical Centre with one haemodialysis department	An exploratory sequential mixed-methods study Questionnaire Observations of nurse–patient encounters using semi-structured observation sheet	30 nurses and 102 patients	Mean age = 65 (SD = 6.3) Male = 59% Average haemodialysis vintage = 60 months (SD = 11.7)	Descriptive statistics, correlational analysis Mixed model linear analysis
Sultan et al. (2022) [18] Egypt	Dialysis centres	Cross-sectional study Interview Laboratory results from patients’ records	205 patients	Mean age = 45.9 years Male = 131 (63.9%) Female = 74 (36.1%) Mean duration on dialysis = 6 years (range 0.2–25 years)	Multivariable logistic regression, McNemar test
Alzahrani et al. (2021) [32] Saudi Arabia	Three haemodialysis centres at three major governmental hospitals	Cross-sectional survey Survey questionnaires Review of patients’ medical files.	361 patients	Mean age = 50.05 years (± 0.83) Male = 47.65% Female = 52.35% Married = 62.33% Unemployed = 321 (88.92%) Personal transportation = 277 (76.73%)	3-stage hierarchical logistic regression analysis
Kim & Cho. (2021) [33] Korea	Online recruitment From three social media communities (kidney patients’ community, nationwide kidney patients’ community in Naver Band, and community for patients with kidney diseases in Daum café)	Descriptive survey Online questionnaire	100	Mean age = 51.70 ± 9.40 Married = 77%, bachelor’s degree or higher = 51% Unemployed = 55%, low economic status = 47% The most common primary cause of ESRD: glomerulonephritis (39%) Mean health status score was 2.92 ± 0.96 Rating their health as “moderate” = 47% Mean haemodialysis duration was 7.57 ± 7.21 Annual average frequency of self-care behaviour education = 8.17 ± 16.27. One educational session per year = 23%, moderate social support = 70% Mean self-care behaviour score = 3.52 ± 0.57 Mean treatment adherence score = 4.01 ± 0.48	Descriptive statistics, Kolmogorov–Smirnov test, *t*-test, one-way analysis of variance, and the Scheffe test Pearson correlational analysis Multiple linear regression analyses Hierarchical regression The Sobel test
Perdana & Yen (2021) [34] Indonesia	Two dialysis units	Cross-sectional study Questionnaires and clinical data	153 patients	Mean age = 50.18 (SD = 12.33) Male = 76 (49.7%) Female = 77 (50.3%) Mean duration of dialysis = 36 months Secondary school graduates = 71 (46.41%) Unemployed = 99 (64.7%) 2x/weekly haemodialysis = 142 (92.8%)	Hierarchical multivariate linear regression analysis
Raashid et al. (2021) [17] Pakistan	Nephrology department in one hospital (tertiary care facility)	Cross-sectional study Face-to-face interviews with structured questionnaires	101 patients	Mean age 51.05 ± 13.80 years Male = 84 (83.17%) Female = 17 (16.83%) Median duration on dialysis = 9 months (range 3–24 months) Frequency of dialysis: 2x/week n = 58 (57.43%), 3x/week n = 43 (42.57%) High school and above = 62 (61.39%)	Binary logistic regression Univariate and multivariate regression analysis
Dantas et al. (2020) [35] Brazil	One dialysis clinic	Cross-sectional study Questionnaires	79 patients	Male = 57% Age = 53.1 ± 12.3 years Median time on haemodialysis = 108 months (89–131.5) Patients with diabetes mellitus = 13.9% Patients with cardiovascular or cerebrovascular disease = 26.6%	Pearson or Spearman’s correlation
Lim et al. (2020) [36] Malaysia	Nine dialysis centres	Cross-sectional study Interview with semi-structured questionnaire Medical records review	218 patients	Mean age = 54.8 ± 12.8 years Males = 116 (53.2%) Females = 102 (46.8%) Mean duration on dialysis = 67.2 months (range 6–272 months) Married = 83.9% At least secondary school graduates = 71.5%	Multiple linear regression
Skoumalova et al. (2020) [37] Slovakia	20 dialysis clinics	Cross-sectional study Questionnaire	479 patients	Mean age = 63.6 (SD = 14.1 years) Dialysis vintage = 5.3 (range 3–36 years) Males: 60.7% Low health literacy = 31.5%, Moderate health literacy = 55.3%, High health literacy = 13.2% of patients	Hierarchical cluster analysis, logistic regression models
Snyder et al. (2020) [46] USA	Three Emory Dialysis facilities (tertiary care facilities)	Retrospective cohort study Electronic medical record data	799 patients	Mean age = 57.1 years Male = 438 (54.8%) Female = 361 (45.2%)	Multivariable logistic regression
Alhawery et al. (2019) [45] Saudi Arabia	A tertiary care dialysis centre	Retrospective case–control study Interviews using questionnaire Medical records for laboratory measures	265 patients	Mean age = 61 ± 18.2 years Males (47.3%); females (52.7%) Dialysis vintage = 3.8 ± 3.3 years	Chi-square test and *t*-test
Indino et al. (2019) [38] Australia	Two dialysis units	Cross-sectional study Questionnaires Electronic medical information system	42 patients	Mean age = 54.4 years Male = 25 (59.5%) Female = 17 (40.5%) Mean duration on dialysis = 3.5 years (SD 2.7 years) Married 40.5% Pensioner 78.6%	Logistic regression Multivariable logistic regression
Opiyo et al. (2019) [39] Africa	Renal clinics and dialysis units at tertiary hospitals	Mixed-methods study (cross-sectional design + in-depth interview) Survey with structured questionnaires, in-depth interviews	333 patients for quantitative, 92 participants (52 patients and 40 family caregivers) for qualitative	For quantitative: Mean age = 46.7 (SD ± 17.3) Male = 199 (59.8%) Female = 134 (40.2%) Married = 221 (66.4%) Secondary school and above graduates = 198 (59.5%) Unemployed = 159 (47.7%) For qualitative: Male = 42 Female = 50 Age range 41–60 years	Univariate and multivariate logistic regression models for quantitative results, thematic analysis, content analysis, and quasi-statistics analysis for qualitative result
Ozen et al. (2019) [40] Turkey	Four haemodialysis centres	Cross-sectional study Face-to-face interviews with questionnaires Patients’ medical records	274 patients	Mean age = 62.57 years ± 13.24 Male = 125 (45.6%) Female = 149 (54.4%) Primary school graduates = 52.6% Married = 79.9%	Multivariable logistic regression analysis Chi-square test
Skoumalova et al. (2019) [41] Slovakia	20 dialysis clinics	Cross-sectional study Questionnaires Medical records	542 patients	Mean age = 63.58 ± 14.12 Male = 329 (60.7%) Female = 213 (39.3%)	Binary logistic regression
Miyata et al. (2018) [42] USA and Japan	Outpatient-based haemodialysis centres	International cross-sectional study Structured interview Patients’ medical records	116 patients in Japan 100 patients in US	Japanese patients’ mean age = 66 years US patients’ mean age = 57 years 81% of Japanese patients completed high school 91% of US patients were high school graduates	*T* tests, Mann–Whitney tests, χ2 tests, linear regression, and logistic regression
Mukakarangwa et al. (2018) [43] Africa	Three referral dialysis centres	Cross-sectional study Structured face-to-face interview	41 patients	Age (range 18 to >60 years) Male = 24 (58%) Female = 17 (42%) Married = 28 (68%) Secondary school graduates = 16 (39%) Unemployed = 31 (75%) Christian = 39 (95%)	Inferential statistics of Chi-square
Washington et al. (2018) [44] USA	Haemodialysis facilities	Cross-sectional observational study Survey	107	Mean age = 63 ± 8.6 Average dialysis vintage = 86 months Male:Female: 51%:49% High school graduate or obtained a GED = 30% Married = 40%, black = 65% Reported poor or fair health status = 54% Lived with others in a private residence = 64% Not depressed = 80%	Chi-square, two-tailed *t*-test, multivariate logistic regression, Hosmer–Lemeshow goodness-of-fit test, likelihood ratio test, and the Breusch-Pagan test

## Data Availability

The data that support the findings of this study are available in Appendix A of this article.

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
