# Peer review of "Factors Contributing to Non-Adherence to Treatment Among Adult Patients with Long-Term Haemodialysis: An Integrative Review"

_nursrep, 2025, doi:10.3390/nursrep15090314_

Round 1
Reviewer 1 Report
Comments and Suggestions for Authors
Dear authors
Thank you for the opportunity to review your manuscript. Below are detailed comments and suggestions based on the feedback, aligned with the PRISMA guidelines and best practices for integrative reviews:
Page MJ, McKenzie JE, Bossuyt PM, Boutron I, Hoffmann TC, Mulrow CD, et al. The PRISMA 2020 statement: an updated guideline for reporting systematic reviews. BMJ 2021;372:n71. doi: 10.1136/bmj.n71

Author Response
Reviewer One
We sincerely thank the reviewers for the constructive comments. Our research team, after discussion and consideration, has revised the manuscript accordingly. We would also like to acknowledge the approval of the extension for this revision. Please kindly find our responses to the comments below:
Reviewer One
Nursing reports -: Factors contributing to non-adherence to treatment
among adult patients with long-term haemodialysis
Dear authors
Thank you for the opportunity to review your manuscript. Below are detailed
comments and suggestions based on the feedback, aligned with the PRISMA
guidelines and best practices for integrative reviews:
Page MJ, McKenzie JE, Bossuyt PM, Boutron I, Hoffmann TC, Mulrow CD, et al.
The PRISMA 2020 statement: an updated guideline for reporting systematic
reviews. BMJ 2021;372:n71. doi: 10.1136/bmj.n71
Response
Thank you for your detailed comments on the manuscript, which we have reviewed as a research team and made changes accordingly. Your comments and our responses have enhanced the manuscript. We have revised the manuscript in line with the PRISMA 2020 statement. Please see the changes made to the manuscript and the responses made to each comment.
Comment 1: Title
The title should identify the report as an integrative review, and the phrase “The text title can be deleted.”
Please revise the title to explicitly state that this is an integrative review, in accordance with PRISMA guidelines.
Response: Thank you for this valuable suggestion. We have revised the manuscript title to explicitly indicate that this study is an integrative review.
Revised Title
Factors contributing to non-adherence to treatment among adult patients with long-term haemodialysis: An integrative review
Comment 2: Abstract
Keywords should be listed in alphabetical order. Kindly reorder the keywords alphabetically to enhance clarity and consistency.
Response: Thank you for pointing that out. We have rearranged the keywords in alphabetical order.
Comment 3: Background
See comment 9 about references
Response: Thank you for the comments about references. We have made changes as stated by the reviewer in comment 9 regarding references.
Cross references to comment 9
We have corrected the referencing errors. As for the old references, we have updated them, except for references 51 and 52. Reference 51 discusses tool validity, and reference 52 explains the “gold standard of measuring non-adherence.” Therefore, they have remained in the manuscript
Comment 4: Methodology
- Section 2.1 is missing a title for “Design.” Add a clear subheading for 2.1 Design
Response: We have added a subheading “Design” at “2.1.”
- Abbreviations are used without a list. Include a list of abbreviations used in
the manuscript. For example (eGFR), CKD
Response: We have added a list of abbreviations above the Appendix section. Appreciate your suggestion.
- Full search strategies (databases, filters, limits) are not presented only search terms and PRISMA. Present the full search strategy for each database, including filters and limits.
Response: We have expanded the search strategy section according to your suggestions. Please kindly find the revision in section 2.2. Thank you.
- Why no updated search? Search until June 2024, why the limitation from the beginning? 1st August 2018 to 30th June 2024.
Response: We have updated the search. The review period has been extended to 30 June 2025.
- The research question should precede the methods. Move the research question to precede the methods section.
Response: Thank you for your suggestion. We have moved the research question.
- Ensure the aim is consistently stated in both the abstract and main text (lines 71–73).
Response: We have amended the aim and objectives in line with the abstract so they are consistent with the main text.
- Data analysis should be more clearly described using Whittemore & Knafl’s framework. Expand the data analysis section to reflect the integrative review method by Whittemore & Knafl.
.
Response: We have amended the description and made revisions to the manuscript.
- JBI is used for cross-sectional appraisal but not referenced for each methodology, please check for example case control study, mixed method, and interviews.
Response: We have created two small tables (one for the JBI cohort study and one for the case-control study). 27 out of 29 included studies are cross-sectional studies (including the ones from mixed-method studies).
- Comment 5: Results
The line numbers from 3.4.1. Social-demographic Factors seem not to be correct You express Fifteen included studies examined the prevalence of non-adherence to at least one component of treatment (fluid, diet, and haemodialysis), and only one study reported … this part is not referenced
Response: Thank you for pointing this out. We have checked and added the reference.
- Table 2 includes mixed-method studies, yet qualitative data were said to be excluded. Clarify how mixed-method studies were handled, especially regarding the exclusion of qualitative data. If qualitative components were excluded, explain the criteria and process. In result you mention 3.2 Data was collected via surveys, structured interviews, extraction of laboratory results, and medical records.
Response: We have only included quantitative results from mixed-method studies. Additionally, we have expanded the inclusion and exclusion section to enhance clarity and understanding.
- Comment 6: Discussion
Ensure the discussion interprets findings considering the research question and existing literature and reflects the integrative nature of the review. Your results are referenced in the discussion for example line 202-203 which need to be checked, your result- then discuss according to other references, no a bit hard to follow as I think the references not are correct.
Response: Thank you for your advice. We have corrected the references to ensure the findings are discussed integratively and allow for results to be compared with previous studies.
- Comment 7: Limitations
Section 4.4 (Limitations) should follow the discussion. Relocate the limitations section to follow the discussion. More limitations should be acknowledged, expand the section to include all relevant limitations, such as cross-sectional data and exclusion of inaccessible articles.
Response: We have relocated the limitations section and added the limitations to follow the discussion. We have also expanded the limitations section as reflected in the revised manuscript.
- Comment 8: Formatting -Line endings should be checked for readability. Please review and adjust line breaks and formatting throughout the manuscript to improve readability and flow.
Response: We have rearranged the formatting line endings.
- Comment 9. References. Check for updates, references from 2010, 2014 may be updated
References are not correct; you refer to WHO line 48 but the reference is
Murali, K. M.; Lonergan, M. Breaking the adherence barriers: Strategies to improve treatment adherence in dialysis patients. Semin. Dial. 2020, 33 (6), 475-485.
https://doi.org/10.1111/sdi.12925 [ProQuest] so the references need to checked out, that makes so the references all over does not seem to be correct. References in tables need author name, to check. I guess references is not correct, please check
Response: We have corrected the referencing errors. As for the old references, we have updated them, except for references 51 and 52. Reference 51 discusses tool validity, and reference 52 explains the “gold standard of measuring non-adherence.” Therefore, they have remained in the manuscript as we feel that they are required.

Reviewer 2 Report
Comments and Suggestions for Authors
This manuscript delivers a comprehensive, methodologically sound, and clinically relevant integrative review on factors contributing to treatment non-adherence in adult patients receiving long-term haemodialysis. It synthesizes findings from 20 studies conducted globally, offering useful thematic categorization of contributing factors (socio-demographic, clinical, and self-management/perception-based). The topic is highly significant given the persistent global burden of end-stage renal disease, and the review appropriately employs Whittemore and Knafl’s framework and the JBI appraisal tool to ensure methodological rigor.
The authors succeed in presenting a logically organized and well-supported discussion that enhances understanding of the multifactorial nature of treatment non-adherence. The conclusions are generally well-aligned with the evidence provided.
While the manuscript is strong and suitable for publication, the following minor revisions are recommended to enhance clarity, transparency, and reader utility.
Strenghts:
Timely and Relevant Topic: Non-adherence remains a leading barrier to improved outcomes in haemodialysis patients, making this review highly valuable to clinicians, educators, and policy-makers.
Robust Methodology: Clear use of an established integrative review framework and appropriate use of PRISMA guidelines.
Clear Thematic Analysis: Grouping factors into three main categories aids in digesting complex findings.
Critical Reflection: The discussion section shows awareness of key limitations, including the lack of standardized adherence definitions and heterogeneous measurement tools.
Minor Recommendations:
1. Clarify Critical Appraisal Results
The manuscript would benefit from a brief table or appendix summarizing how each study scored against the JBI checklist, highlighting which studies had limitations (e.g., lack of validity data, unclear settings) and how that influenced synthesis. If a full table is impractical, at minimum provide a short paragraph clarifying how many studies were rated as high/moderate/low quality.
2. Acknowledge Exclusion of Qualitative Studies
Although the rationale for excluding qualitative studies is explained, the authors might briefly acknowledge the limitations this places on exploring patient perceptions and narratives, and recommend future qualitative synthesis as a complementary approach.
3. Clarify Ambiguous Interpretations
The counterintuitive finding that positive nurse attitudes may correlate with non-adherence should be more clearly explained or contextualized within a theoretical framework (e.g., patient autonomy vs. professional guidance).
4. Add COVID-19 Contextualization
The manuscript includes strong COVID-19-related findings; however, these should be clearly contextualized as temporally limited observations.
Author Response
We sincerely thank the reviewers for the constructive comments. Our research team, after discussion and consideration, has revised the manuscript accordingly. We would also like to acknowledge the approval of the extension for this revision. Please kindly find our responses to the comments below:
Reviewer Two
We thank the reviewer for the comments on our manuscript and suggestions for enhancement.
Comment 1 Clarify Critical Appraisal Results
The manuscript would benefit from a brief table or appendix summarizing how each study scored against the JBI checklist, highlighting which studies had limitations (e.g., lack of validity data, unclear settings) and how that influenced synthesis. If a full table is impractical, at minimum provide a short paragraph clarifying how many studies were rated as high/moderate/low quality.
Response
Thank you for this valuable suggestion. While we acknowledge the importance of summarising study quality, as this is an integrative review with predominantly cross-sectional studies, assigning high, moderate, or low ratings using the JBI checklist may not be appropriate. The checklist guided our appraisal, but due to the inherent limitations of cross-sectional designs, overall ratings could be misleading. Instead, we provided methodological limitations with tables. We have created two small tables (one for the JBI cohort study and one for the case-control study). Twenty-seven out of twenty-nine included studies are cross-sectional studies (including the ones from mixed-method studies).
Comment 2. Acknowledge Exclusion of Qualitative Studies
Although the rationale for excluding qualitative studies is explained, the authors might briefly acknowledge the limitations this places on exploring patient perceptions and narratives, and recommend future qualitative synthesis as a complementary approach.
Response
We have checked and added information in the recommendation section.
Comment 3 - Counterintuitive finding
The counterintuitive finding that positive nurse attitudes may correlate with non-adherence should be more clearly explained or contextualized within a theoretical framework (e.g., patient autonomy vs. professional guidance).
Response
After closely reviewing the article, we have revised the wording. This finding is not necessarily related to issues of autonomy or professional guidance. In addition, there was only one study related to this area, and although not expanded upon in the manuscript, it could be an area for further research.
Comment 4. Add COVID-19 Contextualization
The manuscript includes strong COVID-19-related findings; however, these should be clearly contextualized as temporally limited observations.
Response
We have reviewed and added information related to COVID-19 in the limitations section.

Round 2
Reviewer 1 Report
Comments and Suggestions for Authors
Dear Authors
Thank you for your thorough and thoughtful revisions. You have addressed each of the reviewers' comments, and the manuscript has been significantly improved as a result.
.